# Differences of the Immune Phenotype of Breast Cancer Cells after Ex Vivo Hyperthermia by Warm-Water or Microwave Radiation in a Closed-Loop System Alone or in Combination with Radiotherapy

**DOI:** 10.3390/cancers12051082

**Published:** 2020-04-27

**Authors:** Michael Hader, Deniz Pinar Savcigil, Andreas Rosin, Philipp Ponfick, Stephan Gekle, Martin Wadepohl, Sander Bekeschus, Rainer Fietkau, Benjamin Frey, Eberhard Schlücker, Udo S. Gaipl

**Affiliations:** 1Department of Radiation Oncology, Universitätsklinikum Erlangen, Friedrich-Alexander-Universität Erlangen-Nürnberg (FAU), 91054 Erlangen, Germany; michael.hader@uk-erlangen.de (M.H.); denizpinar.savcigil@gmail.com (D.P.S.); rainer.fietkau@uk-erlangen.de (R.F.); benjamin.frey@uk-erlangen.de (B.F.); 2Chair for Ceramic Materials Engineering, Keylab Glass Technology, University of Bayreuth, 95447 Bayreuth, Germany; andreas.rosin@uni-bayreuth.de (A.R.); Philipp.Ponfick@uni-bayreuth.de (P.P.); 3Biofluid Simulations and Modeling, Fachbereich Physik, University of Bayreuth, 95447 Bayreuth, Germany; stephan.gekle@uni-bayreuth.de; 4Dr. Sennewald Medizintechnik GmbH, 81829 Munich, Germany; martin.wadepohl@sennewald.de; 5ZIK plasmatis, Leibniz Institute for Plasma Science and Technology, Felix-Hausdorff-Str. 2, 17489 Greifswald, Germany; sander.bekeschus@inp-greifswald.de; 6Department of Chemical and Biological Engineering, Institute of Process Machinery and Systems Engineering (iPAT), Friedrich-Alexander-Universität Erlangen-Nürnberg (FAU), 91054 Erlangen, Germany; sl@ipat.uni-erlangen.de

**Keywords:** hyperthermia, microwave-heating, radiotherapy, immunotherapy, breast cancer, immune checkpoint molecules, EGFR, danger signals, immunogenic cancer cell phenotype, multimodal tumor therapies

## Abstract

The treatment of breast cancer by radiotherapy can be complemented by hyperthermia. Little is known about how the immune phenotype of tumor cells is changed thereby, also in terms of a dependence on the heating method. We developed a sterile closed-loop system, using either a warm-water bath or a microwave at 2.45 GHz to examine the impact of ex vivo hyperthermia on cell death, the release of HSP70, and the expression of immune checkpoint molecules (ICMs) on MCF-7 and MDA-MB-231 breast cancer cells by multicolor flow cytometry and ELISA. Heating was performed between 39 and 44 °C. Numerical process simulations identified temperature distributions. Additionally, irradiation with 2 × 5 Gy or 5 × 2 Gy was applied. We observed a release of HSP70 after hyperthermia at all examined temperatures and independently of the heating method, but microwave heating was more effective in cell killing, and microwave heating with and without radiotherapy increased subsequent HSP70 concentrations. Adding hyperthermia to radiotherapy, dynamically or individually, affected the expression of the ICM PD-L1, PD-L2, HVEM, ICOS-L, CD137-L, OX40-L, CD27-L, and EGFR on breast cancer cells. Well-characterized pre-clinical heating systems are mandatory to screen the immune phenotype of tumor cells in clinically relevant settings to define immune matrices for therapy adaption.

## 1. Introduction

Tumors have developed a multitude of mechanisms to evade immune surveillance [1]. Breast cancer is a heterogeneous disease and represents the second most common cause of cancer deaths among women. Most of the breast cancer types are characterized by a low mutational burden and they are immunologically silent [2]. Therefore, only modest response rates to immunotherapies such as immune checkpoint inhibitors (ICIs) are observed. However, just recently, it has been shown that a combination of the ICI pembrolizumab and radiotherapy (RT) is safe and demonstrates encouraging activity in patients with poor-prognosis, metastatic, triple-negative breast cancer [3]. It has become obvious that locally delivered RT can induce systemic immune modulations. Besides the activation of the immune systems following radiation of cancer cells by, e.g., stimulating the release of danger signals, immune suppressive pathways, such as the increased expression of immune checkpoint molecules (ICMs) are also observed [4]. A big challenge in innovative radiation oncology is the identification of patients who do respond well to distinct combinations of radiation protocols with selected immunotherapies [5]. For this, knowledge about the immune phenotype of cancer cells per se and particularly following treatment with, e.g., RT is mandatory.

For breast cancer, most research has focused on the analysis of the expression of the immune suppressive ICM programmed death-ligand 1 (PD-L1) and its receptor PD-1 [6]. Notably, following radiation, breast cancer cells shed microparticles that are characterized by an increased expression of PD-L1 [7]. Kötter et al. demonstrated an increased release of the danger signal HSP70 of breast cancer cells after exposure to radiation. Interestingly, adding a second stressor, namely heat of 41.5 °C, resulted in an even more increased concentration of HSP70 in the breast cancer cell supernatant [8]. One can conclude that, (I) utilizing the immune activation properties of RT for the treatment of breast cancer, additional blocking of immune suppressive molecules such as PD-L1 is necessary, and that (II) hyperthermia (HT) might act as an additional immune stimulator. However, it is not known by now how HT impacts the expression of both immune suppressive and immune stimulating ICMs, either alone or in combination with RT. 

Hyperthermia, i.e., the local heating of tumor tissue to temperatures between 39 and 45 °C for around 60 min, has been proven to enhance radiation and chemotherapeutic effects [9,10]. As just recently summarized by Paulides et al. [11], in particular, microwave and radiofrequency HT devices provide a variety of quality-controlled heating approaches that allow for the treatment of most solid cancers regardless of the size. By these application forms, adding HT to conventional radio(chemo)therapy was shown to be beneficial for the clinical outcome in the context of randomized phase I–III trials. However, one has to carefully consider which tumors can be heated in the clinics. HT of metastatic lesions deep in the body, such as the lung and liver, are not easy to perform by current available HT systems. Two main reasons are the high heat dissipation in these well-perfused tissue entities as well as (respiratory) motion and air, which affects invasive targeted temperature measurement and makes adequate heat delivery difficult, respectively. However, improvements in treatment planning software (segmentation, tissue bioheat properties etc.) and non-invasive real-time temperature monitoring/adaption might bring improvements to achieve the desired temperature elevation even in such tumor entities in the future [11].

It is further accepted that HT impacts the immune system [12,13,14]. For recurrent breast cancer, HT has become a well-accepted therapy when applied in combination with RT [14,15]. Nevertheless, little is known how HT particularly impacts the immunogenicity of breast cancer cells, whether temperature-dependencies do exist, and if heating methods differ from each other. Willerding et al. investigated the temperature characteristics in tumor-bearing rats by three different heating techniques, i.e., lamp, laser, and warm-water bath. One of their key results regarding the different heating techniques was that intratumoral temperature increase and distribution differed significantly from each other, also in dependence of the tumor size [16]. In this context, in clinical trials, HT is predominantly delivered via radiative/microwave applicators [9] with quality assurance guidelines for superficial [17] and interstitial hyperthermia [18]. In contrast, pre-clinical hyperthermia lacks standardized treatment guidelines and systems, and that is why experiments are mostly based on incubators or warm-water baths. Our described model system allows, for the first time, to analyze the impact of microwave heating on the immunogenicity of tumor cells by avoiding the generation of temperature hot spots. For the translation of pre-clinical findings into clinical trials, it is important to use similar settings and parameters as much as possible. Immune checkpoint inhibitors were clinically approved for the treatment of different solid tumor entities, but the response rates are still below 15% [19]. This raises questions of why so many patients do not respond to immunotherapy, and how can standard cancer therapy methods such as RT contribute to increase the number of responders. HT could be integrated in such multimodal settings as an additional immune modulator [13].

To improve the pre-clinical knowledge on how clinically relevant HT impacts the immune phenotype of breast cancer cells and for future clinical integration of HT in multimodal radio-immunotherapy settings we now focused for the first time on detailed analyses of the immune phenotype of breast cancer cells following treatment with HT with different temperatures and application forms alone and in combination with again clinically relevant normo- or hypofractionated RT. Particularly, an improved understanding of the thermal and non-thermal cell inactivation effects depending on the heating method, i.e., warm-water or electromagnetic radiation, is very important because there are various techniques available to perform both superficial, deep, as well as regional or whole-body HT [20].

## 2. Results

The effects of HT on the immune phenotype of breast cancer cells were investigated using an in-house developed closed-loop media flow system with modular and comparable heating devices (see Section 4.1), which can heat the tumor cell suspensions to clinically relevant temperatures by warm-water (CH) or by microwave heating (MH) in the range from 39–44 °C and beyond. To elucidate the efficacy of HT, in the context of combination therapy with RT, the cells were treated either with one of the single therapies or in a combinatory setting, i.e., HT and RT. Focus was given to the analyses of the inactivation efficiency by monitoring the percentage of apoptotic and necrotic tumor cells, and on the analyses of the immunogenic potential of treated tumor cells through the detection of the danger signal heat shock protein 70 (HSP70) and the surface expression of several immune activatory and immune suppressive checkpoint molecules.

### 2.1. Cell Death Induction by Radiotherapy in MCF-7 and MDA-MB-231 Breast Cancer Cell Lines

After RT with normofractionation (normo) or hypofractionation (hypo), the cell death forms of MCF-7 and MDA-MB-231 breast cancer cells were determined day 3 (d3) and day 5 (d5) by AnnexinV/PI-staining. As shown in Figure 1, both normo- and hypofractionation led to significant increases of dead cells, i.e., the sum of apoptotic and primary as well as secondary necrotic cells, in both human breast cancer cell lines. In both cell lines, the percentage of dead cells increased from days 3 to 5, whereas MDA-MB-231 cells showed slightly higher inactivation rates in comparison to MCF-7. The number of apoptotic cells was also higher in MDA-MB-231, especially on day 3, compared to less radiosensitive caspase-3 deficient MCF-7 cells. Notably, the cultivation of both cell lines without any treatment until days 3 and 5 did not negatively affect their viability.

### 2.2. Cell Death Induction by Hyperthermia in MCF-7 and MDA-MB-231 Breast Cancer Cell Lines

Using our modular, in-house developed, closed loop media flow system (see Section 4.1), cell death induction by HT in MCF-7 and MDA-MB-231 breast cancer cell lines at clinically relevant temperatures, i.e., 39, 41, and 44 °C was investigated. The volume flow rate, ${\dot{V}}_{m}$, was kept constant at 2 mL/s, so only the output power of the microwave and the warm water bath temperature were varied. Since HT is usually performed for an effective treatment time of 60 min (in our system, it is 4 h of cyclic heating in total—see Section 4.1 and Section 4.4), we took samples directly after the HT treatment (d0_60′) and distributed the cell suspension into 75 cm^2^ T-flasks for analysis on days 3 (d3) and 5 (d5). Because there is no ex vivo HT system known that can perform either conventional warm-water (CH) or microwave heating (MH) under well-defined and comparable conditions, these two heating methods were of special interest. The latter might reveal findings about the thermal and non-thermal effects of microwave radiation. 

#### No Significant Cell Death Induction by Conventional Warm-Water Heating but by Microwave Heating

Usually, ex vivo HT studies are performed using incubators or warm-water baths. As shown in Figure 2a,b, heating the tumor cell suspension for an effective time of 60 min (d0_0′ to d0_60′) in the self-designed HT system by conventional warm-water bath did not significantly increase the percentage of dead cells. This was independent of the cell line, the clinically relevant treatment temperatures, as well as the time points of cell death analysis, i.e., 60 min after HT (d0_60′), or in the follow up, on days 3 and 5. Radiative heating with frequencies from around 13.56 MHz to 2.45 GHz (spectrum of radio- and microwaves) is widely used in the clinics, since this heating method yields the best power disposition and temperature distribution. However, in vitro heating experiments by microwaves on lab-scale are barely performed due to technical challenges, e.g., online temperature-monitoring of the cell suspension and homogeneous heating. Using our modular HT system, it is now for the first time possible to compare MH with conventional warm-water heating under standardized conditions at same process conditions (see results in Section 2.6). Directly after 60 min of MH (d0_60′), the percentage of dead cells significantly increased with rising treatment temperature in comparison to untreated cells, except at 39 °C in MCF-7 cell line (Figure 2c,d). While at 39 °C and 41 °C the level of dead cells remained in the low double-digit percentage, HT at 44 °C highly significant inactivated both MCF-7 and MDA-MB-231 breast cancer cells. Nevertheless, cell growth within the period of investigation, i.e., until day 5, could be observed by a decreasing percentage of dead cells in total.

### 2.3. Cell Death Induction by Hyperthermia and Radiotherapy in MCF-7 and MDA-MB-231 Breast Cancer Cell Lines

Hyperthermia should always be applied as an additive method in multimodal treatment settings, i.e., hyperthermia with (chemo)radiotherapy. Therefore, we investigated cell death induction by combining clinically relevant HT, performed by either conventional warm-water or by MH, with different irradiation schemes (see Section 4.4). 

#### Conventional Warm-Water Hyperthermia Barely Induces Further Cell Death in Combination with Normo- or Hypofractionated Irradiation, but Microwave Heating Has Additive Cell Killing Effects

Figure 3a,b depicts cell death forms of MCF-7 and MDA-MB-231 breast cancer cell lines on days 3 (d3) and 5 (d5) after combinatory treatment with normo- or hypofractionated irradiation and warm-water based HT. In both cell lines, the percentage of dead cells was independent of the treatment temperature, but hypofractionation was more effective in comparison to normofractionation. Herein, the inactivation rate of MDA-MB-231 cells was slightly higher in comparison to MCF-7 cells. To mimic clinical applied tumor treatment with HT, the microwave HT heating was used again in combination with normo- and hypofractionated irradiation. As shown in Figure 3c,d, hyperthermia at 41 °C and 44 °C in combination with radiotherapy highly significant increased the cell death rate of MCF-7 cells, whereas normo- did not differ much from hypofractionation. In comparison to MCF-7 at 39 °C, MDA-MB-231 already showed highly significant inactivation rates in comparison to the respective untreated controls, whereas at 44 °C, MCF-7 cells were even more inactivated on d3 and 5.

### 2.4. Release of Danger Signal HSP70 in the Supernatant Following Radiotherapy and/or Hyperthermia

Heat shock proteins (HSPs) play a significant role in response to cell stress, such as high temperatures, as they reverse or inhibit denaturation and unfolding of cellular proteins. This makes HSPs important in cell growth and cancer development, but HSPs that are released also have potential clinical uses as biomarkers for disease progression and immunological response. One of the most prominent HSPs is the 72 kDa heavy HSP70, that was quantitatively assessed by sandwich ELISA from supernatant medium of treated and untreated tumor cells.

#### 2.4.1. The Danger Signal HSP70 is Significantly Increased after Radiotherapy

The protein content of HSP70 in the medium after irradiation, standardized to 1 × 10^7^ cells, is shown in Figure 4 for MCF-7 and MDA-MB-231 breast cancer cells on day 3 (d3) and 5 (d5). In the control experiments, a non-significant increase of HSP70 on day 3 was detected, decreasing until day 5. In both cell lines and on both days (d3 and d5), RT significantly led to an increase of HSP70 in the medium, except on day 3 in normofractionated irradiated MCF-7 cells. In MCF-7, the basal level of HSP70 in the medium was higher in comparison to MDA-MB-231, both in control and RT arms.

#### 2.4.2. Significantly Increased Release of Danger Signal HSP70 Directly after Hyperthermia in MCF-7 and MDA-MB-231 Breast Cancer Cell Lines

The results of HSP70 in the supernatant directly after HT (d0_0′ to d0_60′), as well as in the follow up on day 3 (d3) and 5 (d5), is shown in Figure 5. A significantly increase of HSP70 level was found directly after HT at 39 °C and 41 °C in both cell lines and by both heating methods. At 44 °C directly after HT, only CH led to a significant increase of HSP70 for MCF-7, respectively only MH for MDA-MB-231 cell line. In the follow up on day 3 and 5, no significant effect induced by HT in comparison to the respective control experiments (only control of 39 °C and CH shown) was found. Of note is that, at 39, 41, and 44 °C, microwave heated MCF-7 and MDA-MB-231 cells had higher protein levels on day 3 in comparison to CH.

#### 2.4.3. Release of Danger Signal HSP70 after Hyperthermia and Radiotherapy on Day 3 and Day 5

Figure 6 shows the protein content of HSP70 in the medium after combinatory treatment by normo- or hypofractionated irradiation and HT at clinically relevant temperatures, i.e., 39, 41, and 44 °C. As shown in Figure 6a,b for MCF-7, the combination of MH and RT significantly led to an increase of HSP70, on both days 3 and 5 (d3 & d5) at all temperatures. Under CH and RT, only on day 5 at 41 and 44 °C was a significant increase of HSP70 detected. For MDA-MB-231 (Figure 6c,d) in the combinatory setting, using MH and RT resulted in higher protein content on day 3 in comparison to CH as heating method with some significant raises, especially at 39 °C. On day 5, significant increases of HPS70 could be detected, particularly for MH. 

### 2.5. Impact of Radiotherapy and Hyperthermia on the Expression of Immune Checkpoint Molecules

The clinical outcome in cancer treatment has significantly improved in some patient cohorts since immune-checkpoint inhibitors (ICIs) are used in the clinics, like those targeting the PD-1/PD-L1 and CTLA-4 axes. However, only a low double-digit percentage responds as desired. Little is known about how RT modulates the expression of other ICMs and nothing is known about the effects of HT in this context. Therefore, the impact of HT, RT, and a combination of both treatment modalities (RHT) on the expression of prominent ICMs of the immunological synapse (Figure 7) were investigated on MCF-7 and MDA-MB-231 breast cancer cells before and after (day 3 and day 5) the respective treatments. Focus was set on immune suppressive (PD-L1, PD-L2, HVEM), immune stimulatory (CD137-L, Ox40-L, CD27-L), and immune modulatory (ICOS-L) checkpoint molecules, as well as on the expression of one prominent growth factor receptor, namely the epidermal growth factor receptor (EGFR). The antibodies used for the analyses are summarized in Table 1 and the gating and calculation strategies are specified in Appendix A.

#### 2.5.1. Modulation of Immune Checkpoint Molecules on the Tumor Cell Surface after Radiotherapy

As shown in the heatmap of Figure 8, RT alone increased the expression of PD-L1 on MCF-7 cells slightly by around 20% on day 5, whereas on triple negative MDA-MB-231 cells, it was significantly more, i.e., 59% for hypofractionation and 115% for normofractionation. Radiotherapy further increased the expression of PD-L2 on both MCF-7 and MDA-MB-231 cells significantly on day 5, irrespective of normo- or hypofractionated irradiation. Similar effects were observed for the expression of HVEM and ICOS-L. Regarding immune stimulatory ICMs, a more pronounced and significant increased expression was also seen on MDA-MB231 cells on day 5.

Generally, the expression of the examined ICMs is a dynamic event and was dependent on the time of analyses and of the cell line, but less of the fractionation of the radiation. Triple negative MDA-MB-231 had a higher expression level of ICMs following irradiation in comparison to MCF-7 cells. The expression of the EGF receptor was not significantly modulated by irradiation, except for d3 after the normofractionated irradiation of MDA-MB-231.

#### 2.5.2. Expression of Immune Checkpoint Molecules on the Tumor Cell Surface after Hyperthermia

It is nowadays accepted that HT has the potential to modulate an immune response, e.g., by influencing the release of danger signals, as shown in Figure 5 and Figure 6, and under different conditions in [21,22]. We now examined for the first time the expression of ICMs on breast cancer cells after treatment with HT. For this, MCF-7 and MDA-MB-231 breast cancer cells were treated in the closed-loop media flow system at three different and clinically relevant temperatures, i.e., 39, 41, and 44 °C under CH using a warm-water bath or under microwave irradiation at 2.45 GHz (MH) for an effective time of 60 min. To get first hints about the dynamics of the modulation, the expression of the ICMs was again analysed at two different points of time, i.e., three and five days after HT. 

Significant modulations of prominent ICMs are rarely following HT treatment alone (Figure 9). Nevertheless, e.g., PD-L2, was significantly increased on MCF-7 and MDA-MB-231 cells following CH, but in dependence of the temperature. Interestingly, the immune stimulatory molecules OX40-L, CD27-L were only significantly increased on day 3 on MCF-7 cells after CH with 44 °C. In contrast to RT, the expression of several ICMs was even decreased on breast cancer cells afterwards (blue clusters in Figure 9). On balance, HT alone does not strongly impact on ICM expression on breast cancer cells, both in respect to cell line and time. 

#### 2.5.3. Modulation of Immune Suppressive Checkpoint Molecules on the Tumor Cell Surface after Hyperthermia and Radiotherapy

Immune suppressive ICMs (PD-L1, PD-L2 and HVEM) were significantly modulated in their expression on MCF-7 and MDA-MB-231 cells following combinatory treatment, i.e., HT (CH or MH) and irradiation (normo- or hypofractionation) (Figure 10). While the expression of PD-L1 was particularly significantly increased at day 5 after treatment of MCF-7 cells with MH with 41 °C in combination with RT, the impact of RT plus HT on the expression of PD-L2 and HVEM was more rambling. For PD-L2, more clusters of significant upregulation could be identified in comparison to PD-L1, but mostly after combination of CH with RT. However, at day 5, MH at 41 °C in combination with RT resulted in significant increased expression of PD-L2 on both tumor cell lines. The same was observed for increased expression of HVEM on MDA-MB231 cells, but here at temperatures of 39 °C and 41 °C by MH. One can conclude that, generally, the expression of ICMs following HT and RT is individually influenced independent of the tumor cells, temperatures, and the method of heat application (Figure 10).

#### 2.5.4. Modulation of Immune Stimulatory Checkpoint Molecules on the Tumor Cell Surface after Hyperthermia and Radiotherapy

As identified for immune suppressive checkpoint molecules (Figure 10), also the expression of immune stimulatory checkpoint molecules on MCF-7 and MDA-MB-231 breast cancer cells was individually influenced in dependence of the tumor cells, temperature and way of application of heat (Figure 11). Regarding MCF-7 cells, particularly on day 3 after irradiation and HT at 44 °C, a significant upregulation of CD137-L, Ox40-L, CD27-L, and ICOS-L was observed. On day 5 for the same cell line, some clusters of upregulation were found, e.g., for MH at 39 °C and 41 °C for ICOS-L or for Ox40-L at 39 °C and 44 °C. For MDA-MB-231, significant increased expression of immune stimulatory checkpoint molecules was particularly observed at day 5 following RT and CH as wells as for MH at 41 °C, while decreased expression was even observed at day 3 after MH at 44 °C (Figure 11). 

### 2.6. Numerical Simulations to Demonstrate Comparable Heating Conditions of CH and MH

Detailed information about radiation dose (unit Gy) and absorbed energy (unit SAR, respectively °C) distribution to a particular tumor region are mandatory, whereas in the clinics, numerical simulations are required for treatment planning and optimization. To get detailed information about heating conditions under CH or MH inside our modular closed-loop system (Figure 16), numerical simulations of heat transfer and medium flow were performed as described in detail in Section 4.2 by using the commercial software COMSOL Multiphysics^©^ 5.4. Figure 12 illustrates the simulated radial temperature curves at five horizontally aligned intersecting lines for microwave (left, MH) and conventional (right, CH) heating at exemplary process conditions, i.e., *T*_target_ = 44 °C and ${\dot{V}}_{m}$ = 2 mL/s. The wall area of the respective heating unit is filled by gray shadow, the fluid medium is blue colored, starting in the central axis (*r* = 0 mm). For better comparability of the heating units, the results are depicted against normalized length, *L*(*z*)/*L*_0_. In both heating systems, the minimum fluid temperature is in the center (*r* = 0 mm) and the maximum is close to the wall. From the five longitudinal sections (*L*/*L*_0_) one can see that for CH, the temperature close to the central axis is higher in comparison to MH, whereas for the wall area, MH results in higher temperatures especially for *L*/*L*_0_ = 0.5 to 1.0. As shown in Figure 12a, MH results in higher wall, but in lower central axis temperature compared with CH. To quantify this, we performed volume integration of the respective heating unit and evaluated the percentage of the temperature range in Figure 12b. To demonstrate this method for a *T*_target_ of 44 °C, the whole fluid volume reaches more than 37 °C, i.e., 100 % at *T* ≥ 37 °C. For *T* ≥ 41 °C the share decreases to 54.2 % under CH, and 49.5 % under MH. For *T* ≥ 45 °C (*T* > *T*_target_) the share is 0.0 % at CH, it is 9.8 % for *T* ≥ 45 °C, followed by 5.1 % for *T* ≥ 46 °C, 1.8 % for *T* ≥ 47 °C under MH. The reason for this so-called temperature “overshoot” (grey area in Figure 12b) is the factory-made positioning of the stub tuners (see Figure 16c) to enable adequate coupling of the electromagnetic wave into the fluid as well as undisturbed operation. 

## 3. Discussion

### 3.1. Augmenting Breast Cancer Cell Death by Microwave-Based Hyperthermia and Radiotherapy 

It has become obvious that RT, besides its local cytotoxic effects on tumor cells, impacts the immune system. This is the reason why many clinical trials nowadays focus on the combination of RT with immunotherapies [23]. Since HT has also immune modulatory properties, it is reasonable to examine how RT in combination with HT does change the immunogenic phenotype of tumor cells to identify the most beneficial combinations and to gain further knowledge of immune biological modes of action of RHT. This should improve multimodal tumor therapies in the future that also consider the inclusion of additional immune therapies such as immune checkpoint inhibitors (ICIs) [4,12,24,25]. Particularly breast cancer is in the current focus for combination therapies of RT, immune therapy and HT, since strong hints exist that HT and immune therapy with ICIs improve the existing therapies [3,26]. Again, in breast cancer, various dosing regimens of RT are available. So far, recent clinical studies have agreed that hypofractionation (>2 Gy as single dose per fraction) leads to comparable long-term efficacy and even reduced delayed toxic effects compared to normofractionated RT schemes [27,28,29,30]. 

We found a higher rate of inactivation by hypofractionation compared to normofractionated RT at the same total physical dose of 10 Gy, while MDA-MB-231 cells were slightly more sensitive to radiation than MCF-7 cells, which has already been described in another irradiation setting by Kötter et al. [8]. 

While various HT techniques such as warm-water, radiation/microwave, or capacitive heating do exist, there is still controversy about the most favorable heating technique and the best (pre)clinical result. To answer this question in the clinics, quality assurance guidelines for superficial hyperthermia [17] and interstitial hyperthermia [18] have been provided in recent years by ESHO Technical Committee with the participation of senior STM members and members of the Atzelsberg Circle. Beyond that, for example, numerical phantom simulations of microwave or capacitive heating of superficial tissue were performed showing that heating by microwave radiation was better for the target tumor region [31]. Numerous heating systems and treatment protocols are available for preclinical hyperthermia research. However, a lack of guidelines and experimental descriptions lead to a mismatch in the comparability of the results not only within preclinical experiments, but also when translated in clinics. Additionally, knowledge about immunological features of tumor cells following microwave-based HT is scarce, since adequate heating devices need to be developed and characterized for pre-clinical examinations. Werthmöller et al. performed HT on murine B16-F10 melanoma using radiation/microwave technology [32]. These studies gave first hints that microwave-based HT in combination with RT fosters immune cell infiltration into the tumor. However, the complementary in vitro experiments were performed with warm-water-based heating of the tumor cells. 

For the first time, a system (Figure 13) is now available with which in vitro HT experiments can be carried out with either conventional warm-water or microwave heating, both under precisely defined and comparable conditions. We have used this innovative system to study breast cancer cell death forms and immunogenic features as a function of temperature and time, as well as various heating techniques and phenomena, i.e., thermal and non-thermal effects. While thermal effects are attributed to a measurable rise in temperature, non-thermal effects are related to changes in the polarization of molecules, the cell metabolism, and the cell membrane, which are limited in time and space and are sometimes related to “hot spots” [33,34,35]. However, it is important to mention that our cyclic heating approach differs from stationary hyperthermia experiments, e.g., by using a t-flask. Stationary radiative/microwave model systems are very challenging, especially when considering adequate heating without generating local hot-spots. In animal experiments, stationary hyperthermia is easier to perform and to control by using the body’s own physiological heat dissipation.

We found that conventional warm-water heating (CH) in the temperature range of 39–44 °C did increase cell death of human breast cancer cells significantly immediately after 60 min or in the follow-up (d3 and d5). In contrast, microwave-based HT at 2.45 GHz in the modular lab-scale closed loop system significantly increased the percentage of dead tumor cells, especially at 44 °C, while MCF-7 was slightly more heat sensitive in the follow-up regarding necrosis induction compared to MDA-MB-231. Such differences in connection with the heating method were also found for murine B16 melanoma and various cell lines of human non-small cell lung cancer with regard to the viability of the tumor cells, the structure of the cell membrane, cell cycle arrest, and the production of reactive oxygen species [36,37]. 

As already mentioned, there have been discussions about “non-thermal” effects and “hot spots” due to electromagnetic radiation for many decades [33]. In order to refute or prove that possible “non-thermal” effects of radiative heating could be related to the “overshoot range” also being present in our system (Figure 12), we examined a different heating scheme. While the input power for MH and for CH were kept constant, the base circulation temperature was reduced from 37 °C to 30 °C (Figure 13a). The Δ*T* was fixed to +7 °C, but at a temperature level that is not expected affecting the viability of the cells at a significant level. Interestingly, despite a physiological target temperature of 37 °C, MH was again significantly more effective in killing the tumor cells (Figure 13b), but not as much as it was for *T*_target_ = 44 °C. In the follow up, the percentage of inactivated cells under CH treatment decreased, whereas a higher proportion of inactivated cells was found for MH. This suggests that “hot spot” independent non-thermal effects on tumor cells occur in our setup under MH.

In several countries, HT in addition to chemotherapy and/or RT is only reimbursed in the context of clinical studies for certain tumor entities, e.g., for high-risk soft tissue sarcoma or recurrent breast cancer [9]. In this context, radiative HT is the only technique approved by the FDA, since it enables controlled and adequate heat release with non-invasive use, since unlike CH, MH does not require contact between the heat source and the target [38,39,40]. However, the controlled heat release by MH is still very sensitive to biological variables such as the heterogeneity of the tissue properties and the geometry of the blood flow [31,41,42]. When looking at the data of a combination of RT with HT, RT was the dominant induction source for breast cancer cell death when RT was combined with CH. However, microwave-based heating at 44 °C and radiotherapy was the most common induction source for tumor cell death.

### 3.2. Different Temperature Ranges of Hyperthermia Might be Most Beneficial for Immunomodulation 

RT is used primarily for local tumor control and tumor shrinkage, while its immune activating and immune suppressive effects are currently being investigated in several clinical studies as well as intensively pre-clinically. The focus is mainly set on distinct RT fractionation schemes and on timing of the applications [23,43]. The immunogenic potential of HT is promising but has not been investigated at a comparable level to that of RT [13]. While there is an accepted dose parameter scheme for RT planning, e.g., the clinical target volume (CTV) [44], it is not yet available for HT. A good parameter for the heat dose in HT, however, is proposed and described in a review by van Rhoon [45] as a concept of the cumulative equivalent minutes (CEM43: *T*_90_ of 43 °C [46]). Another good thermal dosing concept is TRISE, which also includes temperature and heating time, but multiplies the sum of all treatments where *T*_50_ rises above 37 °C and standardized it to the total planned time, i.e., 450 min [47]. However, both heating concepts do not deal with questions about the biological mechanisms and the thermal dose effect. In this context, HT is sometimes performed once or twice a week [48], the interval between radiation therapy and HT is not precisely defined [49,50] and not all patients receive the same total number of sessions. For example, in the HYCAN-trial (NCT02369939) for anal carcinoma, a detailed longitudinal immunophenotyping of the patients is performed [51]. First results give hints that when HT is added to radiochemotherapy, particularly cells of the innate immune system such as dendritic cells recover faster in the peripheral blood. However, these immunological data in combination with clinical outcome should be linked to measured temperature profiles of each patient individually. Our data point out that thermal dosing concepts should also take heating technology into account, while the cell inactivation rate is higher for microwave heating starting at a specific threshold temperature. Often relevant parameters are not listed [52,53,54] that makes interpretation of pre-clinical data difficult. Best practice methods should include a physical and technical description of the system(s) and simulations of the temperature distribution in order to avoid, for example “hot spots” or other heating abnormalities. 

To get additional hints about the immunogenic features of breast cancer cells following HT and RHT, we analyzed HSP70 as immune activating danger signal [21] in the tumor cell supernatant. We revealed for the first time that HSP70 was released by the breast cancer cells immediately after 60 min of HT at all examined temperature levels and almost regardless of the heating method. This emphasizes the positive effects of HT immediately after the first treatment, even though the tumor was not heated to *T*_target_, i.e., CEM43.

In the follow-up, particularly three days after the treatment, microwave-based HT with and without radiation therapy led to higher HSP70 concentrations in the tumor cell supernatant. Electromagnetic radiation was shown to increase unfolding of proteins [55] and might thereby induce the release of chaperones such as HSP70. This could also be of great relevance for the clinics, since at days of HSP70 release, an immune stimulatory micro-environment is present following HT treatment that should be utilized, e.g., for activation of immune cells that were attracted inside the tumor by RT [56].

### 3.3. Adding Hyperthermia to Radiotherapy Dynamically and Individually Affects the Expression of Immune Checkpoint Molecules on Breast Cancer Cells 

Immune checkpoint molecules play a central role in the immunogenic potential of tumor cells because they are associated with many mechanisms of immune evasion [57]. The PD-L1/PD1 axis, e.g., is an essential regulator of T-cell activation. Tumor cells make use of it to dampen T cell-mediated anti-tumor immune responses [58]. Targeting immune suppressive ICMs to enhance the organism’s anti-tumor immune response has emerged as a promising immune therapy strategy in a subset of cancer patients, either as single treatment or mostly in combination with standard therapies such as RT [59]. 

Even though it is today well accepted that RT impacts the expression of PD-L1 [60,61,62], little is known about the expression of other ICMs after exposing tumor cells to RT, and nothing has been known about how HT impacts expression of ICMs either as single treatment or in combination with RT [13]. We confirmed in human breast cancer tumor cells that RT increases the expression of PD-L1 on the tumor cell surface in a time- and cell line-dependent manner. However, PD-L2 was significantly up-regulated on both breast cancer cell lines after normo- and hypofractionated RT which stresses that immune matrices should be followed up instead of focusing on one distinct ICM for screening [63]. Biological varieties in tumor tissue are common [64,65,66] and should also be monitored in detail during multimodal therapies. Furthermore, given the binding properties of PD-L1 and PD-L2, the significant upregulation of PD-L2 by tumor cells could compensate for the inhibition of PD-1 signaling when PD-L1 inhibitors are solely used for therapy. In contrast to RT, HT did not strongly influence the expression of ICMs on the tumor cell surface when being applied as a single treatment modality. In the case of MH, only a temperature of 44 °C resulted in significant alterations of distinct ICMs. This result again underlines that both the temperature and the manner in which the tumor tissue is heated influence the immunogenicity of a distinct tumor.

However, in clinically relevant applications, namely combination of RT and HT, particularly combination of MH to 41 °C with RT resulted in significantly increased expression of immune suppressive ICMs (PD-L1, PD-L2 and HVEM) on day 5 after treatment. This highlights the dynamics of the changes as already being observed for PD-L1 expression and immune infiltration early during treatment of melanoma patients with anti-PD-1 therapy [67]. One has to emphasize that even temperatures of 39 °C affected the expression of ICMs on breast cancer cells when HT was combined with RT. This again calls for temperature monitoring of the tumor during clinical HT treatment to get hints about changes of the immune phenotype of tumor cells during and after RHT. Changes of the tumor tissue during therapy are of great importance for prognosis, prediction and therapy-adaption and need to be investigated in much more detail and standardized in clinical practice [68,69]. 

However, not only immune suppressive checkpoint molecules were increased in their expression after RHT, as immune stimulatory ones, such as CD137-L, Ox40-L, CD27-L, and ICOS-L, were individually influenced independently of the tumor cells, temperature, and means of HT application. This encourages the early and repeated analysis of tumor tissue or related accessible biomaterial of every single patients for therapy designs and adaptions. Detailed information about the immunological status of the tumor tissue will allow prediction about which patient will benefit most from combined treatments of RT, HT and immunotherapy with selected ICIs [13]. 

In pre-clinical settings, the combined IT of agonistic anti-CD137 antibodies with anti-PD1 or -CTLA-4 antibodies have brought positive outcomes for distinct cancer types [70]. The importance of the knowledge about the concurrent expression of immune stimulatory and immune suppressive ICMs on individual tumor tissue should be shown with the following example: CD137-L was significantly increased on MDA-MB-231 cells after normo-fractionated RT in combination with MH at 39 °C or 41 °C on day 3 after treatment, while for MCF-7 cells, MH heating at 44 °C was required. Under these conditions for MDA-MB-231 and MCF-7 cells, an additional targeting of HVEM would be beneficial. 

For a long-lasting tumor specific protection, OX40 signaling is crucial for the expansion, survival, memory formation and effector functions of the immune system. On the basis of this information, OX40-L fusion protein agonists are currently tested in pre-clinical settings [71]. Based on the same thoughts as outlined above, RHT could increase the expression of OX40-L under distinct conditions and might be combined with OX40 immunotherapy for achieving specific and long-lasting anti-tumor immune responses.

In addition to ICM, the surface expression of the EGFR was examined in addition in the closed-loop flow system for heat treatment strategies of tumor cells with CH or MH alone and in combination with RT. EGFR is overexpressed in many solid tumors including breast cancer and is associated with increased cell growth, angiogenesis, and blocking of cell death [72]. The metabolic reprogramming mediated by the aberrant expression of EGFR, which can improve the pentose phosphate pathway and aerobic glycolysis to promote DNA repair and apoptosis resistance, is an efficient oncogenic defense mechanism of tumor cells [73,74]. Our studies show for the first time, that besides affecting the immunogenic potential of breast cancer cells, MH also impacts EGFR expression, again independent of temperature, way of heating, RT fractionation, and time. It is becoming more and more evident that joint targeting of oncogenic pathways and diverse immune evasion mechanisms enhance anti-tumor responses [75].

Well-characterized pre-clinical heating systems, such as the one presented in this paper, are mandatory to screen tumor cells for their immunogenic and oncogenic features in clinically relevant settings. Additionally, our closed-loop system could be used in the future to generate whole tumor cell-based vaccines based on the concept that was just recently described by Seitz et al. [76]. Detailed information about the phenotype of tumor cells during RHT will result in improved designs of clinical studies for multimodal treatments of breast cancer in the future.

## 4. Materials and Methods

### 4.1. Closed-Loop System for Heat Treatments of Tumor Cells

Ex vivo heat treatment of the tumor cell suspension was performed in a self-designed, modular lab-scale closed-loop media flow system under sterile conditions (Figure 14). Besides electropolished stainless steel (V4A) and quartz glass tubes, a silicone hose (Watson Marlow, platinum-cured silicone tubing, 4.8 mm bore × 1.6 mm wall) was used for the peristaltic pump (Drifton LABF6/2*YZ151X-PPS, Shenzhen, China). During the periodic heating, the base temperature was set to 37 °C by using a cooling bath (Colora WK3DS, Lorch, Germany). The room temperature was kept constant at 20 °C. The closed-loop system was designed for target temperatures (*T*_target_) in the range from 37 °C to 60 °C at flow rates between 2 mL/s and 10 mL/s. To achieve *T*_Target_, either a warm-water bath (CH) or a microwave heating unit (MH) at 2.45 GHz was used. Within the closed loop system, temperature was monitored at the inlet (TIR01) and outlet (TIR02) of the respective heating unit as well as within the cooler (TIR03) and at the end of the variable length (TIR04). Furthermore, temperature was monitored for the CH bath (TIR05) respectively the level, i.e., power, of the microwave unit.

The conventional warm-water device was a self-designed stainless-steel loop (d_i_ = 4 mm) within a temperature-controlled heating bath (Julabo Labortechnik GmbH, Model EC-5, Seelbach, Germany). The microwave setup consisted of a 2 kW-microwave generator (TM A20 S1, Richardson Electronics Ltd.) with a circulator for catching and dissipating the reflected power. A rectangular waveguide (WR-340, 86 mm × 43 mm) was used with two horizontally aligned quartz tubes (d_i_ = 10.2 mm, tube length 8.6 mm) at the applicator’s closed end, and with manual stub tuners for fine-tuning the energy absorption. The media volume in both heaters, *V*_1_, was kept constant at 6.88 mm^3^. V_2_ was set to 25 mm^3^ and V_3_ was 8.50 mm^3^. The total system volume, *V_m_*, was fixed to 129 mm^3^ so V_4+Δ_ was 88.62 mm^3^. 

The flow rate dependent temperature cycle with its characteristic points of time, *t*_i_, is illustrated schematically for one heating cycle in Figure 15a and for periodic heating at two different flow rates in Figure 15b with identical effective treatment time. 

The effective treatment time, *t*_eff_, can be expressed by the amount of cycles, *N*_0_, and the heating time per cycle, $\bar{t}$_heat_. The latter can also be described by the duration of one cycle, t_cycle_, and the relation between the sum of the heater subvolume, *V*_1_, and the connecting pipe volume, *V*_2_, and the total medium volume, V_m_, according to Equation (1):(1)$$t_{\mathrm{eff}}=N_{0}\cdot {\bar{t}}_{\mathrm{heat}}=N_{0}\frac{V_{1}+V_{2}}{V_{m}}\cdot t_{\mathrm{cycle}},$$

Within the experiments, the parameter levels of the effective treatment time, *t*_eff_, and the target temperature, *T*_target_, were varied at a fixed volume flow rate, ${\dot{V}}_{m},$ of 2.0 mL/s (see Section 4.4). Therefore, the time of one cycle, *t_cycle_*, was around 64.5 s (*V_m_*/${\dot{V}}_{m}$). According to Equation (1), the amount of cycles,$\;N_{0}$, is 223 for ${\dot{V}}_{m}$ = 2.0 mL/s and *t_eff_* = 60 min. In our system setup, the effective heating time is ¼ of the total cyclic heating time, i.e., 60 min of effective heating is 4 h of cyclic heating in total.

### 4.2. Numerical Simulations and Modeling of the Heating System

The real temperature distribution within the heating device is unknown as the temperature of the cell suspension can only be measured at the inlet and outlet (Figure 14). Therefore, numerical simulations with the commercial software COMSOL Multiphysics^©^ 5.4. (COMSOL Multiphysics GmbH, Göttingen, Germany) were conducted to achieve the theoretical temperature profiles of the two heating devices. The numerical calculation of electromagnetic field distribution, heat transfer, and fluid flow is based on the finite element method. Maxwell’s equations in the frequency domain resulted in electromagnetic energy distribution, respectively electric and magnetic field strength, E and H, at all points in the computational domain. Equation (2) describes the dissipated electromagnetic power loss density, *P*_loss_, that describes the volume-specific conversion of electromagnetic energy into heat due to dielectric heating, wherein *f* is the electromagnetic wave frequency, $\varepsilon_{0}$ the permittivity of vacuum, $\varepsilon_{r}^{\prime \prime }$ the relative dielectric loss factor and **E** the electric field strength.
(2)$$P_{\mathrm{loss}}=2\pi f\varepsilon_{0}\varepsilon_{r}^{\prime \prime }{\left|E\right|}^{2},$$

Within the heat transfer Equation (3), the dissipated electromagnetic power loss density, *P*_loss_, was used as heat source term, wherein$\;\rho_{m}$ is the density,$\;c_{p}$ the specific heat capacity and $\lambda_{m}$ the heat conductivity of the medium. Navier–Stokes equations for fluid flow conditions were additionally solved.
(3)$$\rho_{m}c_{p}\frac{\partial T}{\partial t}=\nabla \left(\lambda_{m}\nabla T\right)+P_{\mathrm{loss}}$$

A temperature change in fluids alters their materials properties, and the other way around it influences the flow field. Thus, a two-way coupling between fluid flow and heat transfer, i.e., non-isothermal flow physic, was used in COMSOL Multiphysics© simulations. At the outer surfaces heat loss by free convection was used as boundary condition. Furthermore, because of unknown cell suspension characteristics, pure water was used instead. COMSOL Multiphysics© provides an internal database for pure water which was expanded by temperature-dependent complex permittivity $\varepsilon_{r}$ extracted from [77] as shown in Table 2. 

A summary of the temperature-invariant material properties for the microwave (MH) and warm-water bath (CH) models is shown in Table 3.

The regions of interest for the numerical simulations within the modular closed-loop system are highlighted in Figure 16a and the respective 3D simulation models of the warm-water bath (CH) loop as well as the microwave applicator (MH) are presented in Figure 16b,c. As boundary conditions in both heating systems, laminar non-isothermal fluid flow was assumed and relative pressure, *p*_out_, was set to zero Pa at the outlet.

To achieve a homogenous heating in the desired process range (Table 4), the CH loop has an inner diameter of 2 mm and a wall thickness of 1 mm (Figure 16b). All used material properties are shown in Table 3. At the inlet of the loop, the cell medium temperature was 37 °C, and the heat transfer of the warm-water bath was adjusted to achieve *T*_target_. The latter was simulated by convective heat flux, more precisely with external forced convection and cylinder in cross flow settings. Meshing of the CH model was physics-controlled with element size *fine*, and thus resulted in a total of 159932 elements. 

The 3D simulation model of the microwave applicator is presented in Figure 16c. The rectangular metal resonator was regarded as a perfect electronic conductor. To achieve enough microwave absorption to the fluid-filled tubes, metal stub tuners protrude into the air-filled waveguide and can be altered in height. The front water tube was used with 30 °C warm-water at a flow rate of 10 mL/s and acted as a dummy load to catch some of the microwave energy. The rear tube contained the cell medium and the inlet temperature was fixed to 37 °C. To reach the target temperature by adapting the microwave power, *P*_mw_, an electromagnetic wave at 2.45 GHz was defined for excitation in TE10 mode at the rectangular microwave port. The temperature profile of the fluid was re-calculated iteratively since the permittivity and power absorption are mutually dependent, i.e., in a first run the electromagnetic field was calculated followed by a simulation of the fluid and temperature profile with updated local permittivity distribution. This calculation procedure was done several times to achieve a stationary electromagnetic field as well as temperature distribution. To guarantee sufficient mesh accuracy without extended calculation time, a user-controlled mesh was used with corner refinement, boundary layers, free tetrahedral and size adaption. The total number of elements of the microwave model was 198803. 

### 4.3. Cell Lines and Cultivation

The cell lines were cultivated at 37 °C in 5% CO_2_ and 90% humidity under sterile conditions. The two human breast cancer cell lines MCF-7 (MCF-7; Merck KGaA, Darmstadt, Germany) and MDA-MB-231 (MDA-MB-231; Merck KGaA, Darmstadt, Germany) were grown in Dulbecco’s modified Eagle’s medium (DMEM; PAN-Biotech GmbH, Aidenbach, Germany) supplemented with 10% fetal bovine serum (FBS; Biochrome AG, Berlin, Germany), 1% sodium pyruvate, 2 mM glutamine, 100 U/mL of penicillin and 100 µg/mL. MCF-7 cells (ER positive; PgR positive; p53wt) are deficient of caspase-3, while MDA-MB-231 are triple negative, i.e., ER negative; PgR negtaive; p53 mutated, and caspase-3 intact [78,79,80,81]. Both splitting and harvesting the cells was conducted by trypsination (10% trypsin in DPBS) on a heating plate.

### 4.4. Treatments and Sampling

One day prior to the respective treatment (d_-1d_), the cells were seeded in accordance with the particular cell line, treatment and the sampling day, so that the confluence never exceeded 90%, e.g., 3 × 10^5^ cells per 75 cm^2^ t-flask of MDA-MB-231 for sampling on day 5 in the control arm. In the HT and combinatory arm, i.e., RT & HT (RHT), sample injection of 1 × 10^7^ cells into the HT system was done at time point d0_0′, followed by an effective heating session of at most 60 min (d0_60′). During the (R)HT session at a constant flow rate of $\dot{V}$_m_ = 2.0 mL/s under CH or MH at *T*_target_ 39 °C, 41 °C or 44 °C, optional sampling at time points 10′, 20′ and 30′ could be done. It has to be considered that the effective heating time is ¼ of the total cyclic heating time, i.e., 60 min of effective heating is 4 h of cyclic heating in total. After the HT treatment within the closed-loops system (Section 4.1 and Table 4), cells were splited into 75 cm^2^ t-flasks according to their subsequent treatment. Cells in the RHT arms were stored in the incubator for 2 h after HT treatment and subsequently irradiated, as it is common in the clinical applications.

Irradiation in the RT arm (RT) and combinatory RHT arm was performed with either normofractionation with a single dose per fraction of 2 Gy of X-ray (120 kV, 12.2 mA for 0.5 min), or hypofractionation with a single dose per fraction of 5 Gy (120 kV, 21.5 mA for 0.7 min) using an X-ray system (Cabin: Seifert; Generator: Isovolt Titan series – GE Technologies; Hürth; Germany) and 75 cm^2^ t-flasks. Irradiation in the respective RT and RHT arms was always performed at the same time, with a time interval of 2 h after HT. This is translated from clinical hyperthermia guidelines as the time interval between RT and HT should be around 1 h and less than 4 h [17]. From the biological point of view, the sequence between radiation and heat is very important; the thermal enhancement ratio (TER) is greatest when heat and radiation are given simultaneously. The tumor TER is almost the same, when HT is applied before or after RT, but the time interval has to be less than 4 h [82]. Figure 17 summarizes the experimental set-up.

Standard sampling in all treatment arms was performed on day 0 (d0), d3 (72 h) and d5 (120 h). In the HT and RHT arm sample injection of 1 × 10^7^ cells into the HT system was done at point d0, followed by an effective heating session of at most 60 min. Heating under CH or MH was done at *T*_target_ 39 °C, 41 °C or 44 °C. During the (R)HT session, optional sampling at time points 10′, 20′ and 30′ could be done. After the HT treatment, cells were recultivated into 75 cm^2^ t-flasks according to their subsequent treatment, i.e., HT alone or additional RT (RHT). Irradiation in the RT arm and combinatory RHT arm, with a time interval of 2 h to preceded hyperthermia, was performed in 75 cm^2^ t-flasks with clinically relevant doses of either 2 × 5 Gy or 5 × 2 Gy. To be considered, irradiation in the respective RT and RHT arms was always performed at the same time. The samples were collected for further analyses (see Section 4.5, Section 4.6 and Section 4.7).

### 4.5. Cell Death Detection by AnnexinV/PI Staining

Multicolor flow cytometry (FACS Coulter^®^ EPICS^®^ XL^™^ Flow Cytometer; Coulter, Fullerton, CA, USA) of Annexin V/ propidium-iodide (PI) stained cells was used to detect death forms of tumor cells after irradiation and/or HT treatment (Figure 18). Characteristic morphological changes during cell death can already be detected by forward scatter (FSc)/side scatter (SSc) changes. Staining cells with propidium-iodide (PI) allows to identify necrotic cells, since PI penetrates only into cells that have lost their membrane integrity. Due to the latter binding mechanism of PI with DNA, primary necrosis and secondary necrosis can be differed by the intensity of the PI signal [83]. By co-staining with FITC-conjugated AnnexinV as described in [84], apoptotic cells that undergo controlled cell death can be differentiated from primary and secondary necrotic as they express phosphatidylserine (PS) on the outer membrane leaflet but still show an intact membrane. AnnexinV/PI-staining was always performed in duplicates according to the protocol: 100,000 cells/tube resuspended in Ringer (B. Braun, Melsungen, Germany) stained with 1 µg/mL of PI (0.4 μL, Sigma Aldrich, Munich, Germany) and 0.5 µg/mL of FITC-labeled AnnexinV (0.2 μL, Geneart, life technologies, Regensburg, Germany). 

### 4.6. Detection of Heat Shock Protein 70 (HSP70) by ELISA

The amount of heat shock protein 70 (HSP70) in the tumor cell supernatant was analyzed using a sandwich DuoSet^®^ IC ELISA kit (R&D Systems, DYC 1663-5; Minneapolis; MN, USA) as described in manufacturer’s instruction, except an adaption of HSP70 calibrator concentration in the range of 0 ng/mL to 80 ng/mL. Cell culture supernatant for HSP70 ELISA was centrifuged after the respective treatment at 300× g for 5 min at room temperature and stored at −80 °C. 

### 4.7. Detection of Immune Checkpoint Molecule and EGFR Expression by Multicolor Flow Cytometry

The tumor cells were harvested and 1 × 10^5^ breast cancer cells per 96-well were incubated with 100 µL of antibody staining solution (Table 1) for 30 min in the dark at 4 °C. Mastermix #1 contained antibodies against three ICMs, namely PD-L1, PD-L2 and ICOS-L and one against epidermal growth factor receptor (EGFR). Mastermix #2 included antibodies against four ICMs, namely HVEM, OX40-L, TNFRSF9 and CD27-L. All antibody concentrations were tested according to manufacturer’s instruction and finally titrated for both cell lines at a concentration of 0.5 µL/well, respectively 1 × 10^5^ tumor cells. To distinguish viable from dead cells, Zombie NIR at a concentration of 0.1 µl/well was used. The mean fluorescence intensity of stained samples was subtracted from unstained mock-treated samples that contained only FACS-buffer and Zombie NIR. Appendix A provides more detailed information about gating strategy and fluorescence detection. The samples were measured by multi-color flow cytometry (Beckman, Cytoflex S, Krefeld, Germany). The data are presented as normalized expression: change in mean fluorescence intensity compared to mock-treated cells.

### 4.8. Statistical Analysis

The arithmetic means of replicates, as calculated by flow analysis software Kaluza 2.0 (Beckman Coulter; Brea; USA), is depicted. The software Prism 7 and 8 (Graph Pad; San Diego, CA, USA) was used for statistics. For analysis, Kruskal-Wallis (1-way ANOVA) was used. Results were considered statistically significant for * *p* < 0.1, ** *p* < 0.01, and *** *p* < 0.001.

## 5. Conclusions

We identified in comparative tests on closed-loop cell media treatment that HT does not only match to RT because of its radiosensitising properties, but also because of its influence on the immune phenotype of breast cancer cells. Both normofractionated and hypofractionated RT were well combinable with HT in this regard. It is particularly worth highlighting that, independent of the heating source, particular immune phenotypes of breast cancer cells do result. This stresses that heating systems for pre-clinical research should be adapted to the parameters that are used in clinical settings, such as tumor heating by microwave applicators [85]. The developed sterile closed-loop heating system allowed detailed analyses of the influence of heat and microwaves on the phenotype of tumor cells. Our results suggest that non-thermal factors contribute to breast cancer cell killing and also individually shape the immunogenic properties of the tumor cells. Additionally, temperatures of 39, 41, and 44 °C have particular immune modulatory properties, mostly when being combined with RT. While MH with 44 °C increased the necrosis of breast cancer cells, particularly MH with 41 °C in combination with RT affected the expression of ICMs. The early release of the danger signal HSP70 was promoted by MH at all examined temperatures. ICMs, both immune suppressive and immune stimulatory ones, were individually influenced in dependence of the tumor cells, temperature and way of application of HT. Therefore, repeated analyses of tumor tissue and related biomaterial of patients should be performed closely for the improvement of multimodal tumor therapies consisting of radio(chemo)therapy, immune therapy, and hyperthermia [12].

## Figures and Tables

**Figure 1 cancers-12-01082-f001:**
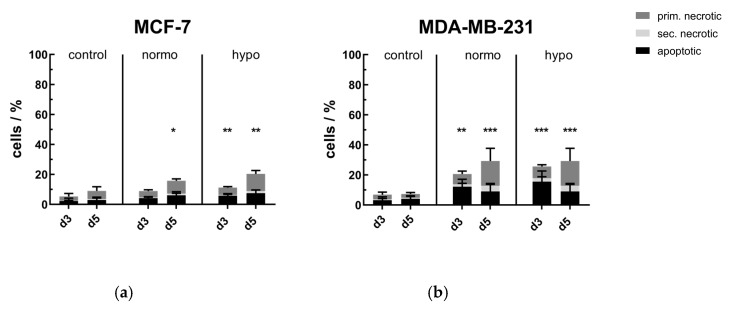
Cell death forms of (**a**) MCF-7 and (**b**) MDA-MB-231 breast cancer cells after normo- and hypofractionated radiotherapy. Cell death forms were analyzed by AxV/PI-staining and multicolor flow cytometry measurement. AxV^+^/PI^-^ cells are apoptotic ones, AxV^+^/PI^++^ cells are primary and AxV^+^/PI^+^ are secondary necrotic ones. The total percentage of dead cells yields the tumor cell killing efficiency. Mean ± S.D. are presented from at least four independent experiments, each measured in duplicates. Significance test was conducted using Kruskal-Wallis test with uncorrected Dunn’s multiple comparison, by comparing the treatment-related total percentages of killed cells to the corresponding controls of mock-treated cells at the indicated time points (d3,d5); * (*p* < 0.1), ** (*p* < 0.01), *** (*p* < 0.001).

**Figure 2 cancers-12-01082-f002:**
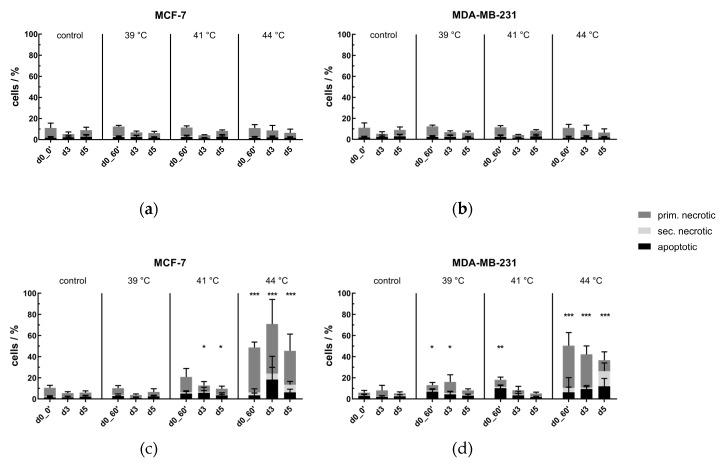
Cell death forms of (**a**) MCF-7 and (**b**) MDA-MB-231 breast cancer cells after conventional heating, or after microwave heating of (**c**) MCF-7 and (**d**) MDA-MB-231 cells. Cells were heated either by conventional warm-water (CH) or microwaves (MH) within the self-designed HT system to three clinically relevant temperatures, i.e., 39 °C, 41 °C and 44 °C, for an effective time of 60 min (d0_0′ to d0_60′). After the effective treatment time of 60 min, the cells were distributed into 75 cm^2^ T-flasks for analysis on day 3 (d3) and day 5 (d5). Cell death forms were analyzed by AxV/PI-staining and multicolor flow cytometry measurement. AxV^+^/PI^-^ cells are apoptotic ones, AxV^+^/PI^++^ cells are primary and AxV^+^/PI^+^ are secondary necrotic ones. The total percentage of dead cells yields the tumor cell killing efficiency. Mean ± S.D. are presented from at least four independent experiments, each measured in duplicates. Significance test was conducted using Kruskal-Wallis test with uncorrected Dunn’s multiple comparison, by comparing the treatment-related total percentages of killed cells to the corresponding controls of mock-treated cells at the indicated time points (d0_0′, d3, d5); * (*p* < 0.1), ** (*p* < 0.01), *** (*p* < 0.001).

**Figure 3 cancers-12-01082-f003:**
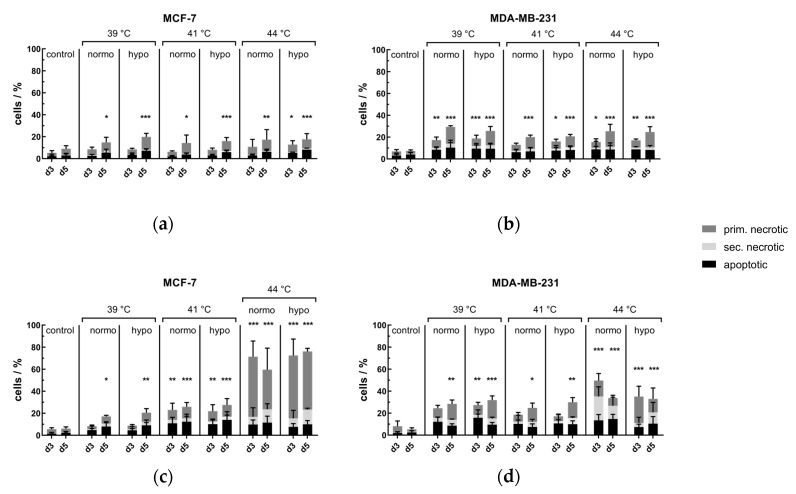
Cell death forms of (**a**) MCF-7 and (**b**) MDA-MB-231 breast cancer cells after combinatory treatment with radiotherapy and conventional heating or after microwave heating of (**c**) MCF-7 and (**d**) MDA-MB-231 cells. Cells were heated either by conventional warm-water (CH) or microwave heating (MH) within the self-designed hyperthermia system to three clinically relevant temperatures, i.e., 39 °C, 41 °C and 44 °C, for an effective time of 60 min (d0_0′ to d0_60). After the effective treatment time of 60 min, the cells were distributed into 75 cm^2^ T-flasks for additional treatment, i.e., normofractionation (normo) at single doses of 2 Gy or by hypofractionation (hypo) at single doses of 5 Gy, and analysis on day 3 (d3) and day 5 (d5). Cell death forms were analyzed by AxV/Pi-staining and multicolor flow cytometry measurement. AxV^+^/PI^-^ cells are apoptotic ones, AxV^+^/PI^++^ cells are primary and AxV^+^/PI^+^ are secondary necrotic ones. The total percentage of dead cells yields the tumor cell killing efficiency. Mean ± S.D. are presented from at least four independent experiments, each measured in duplicates. Significance test was conducted using Kruskal-Wallis test with uncorrected Dunn´s multiple comparison, by comparing the treatment-related total percentages of killed cells to the corresponding controls of mock-treated cells at the indicated time points (d3, d5); * (*p* < 0.1), ** (*p* < 0.01), *** (*p* < 0.001).

**Figure 4 cancers-12-01082-f004:**
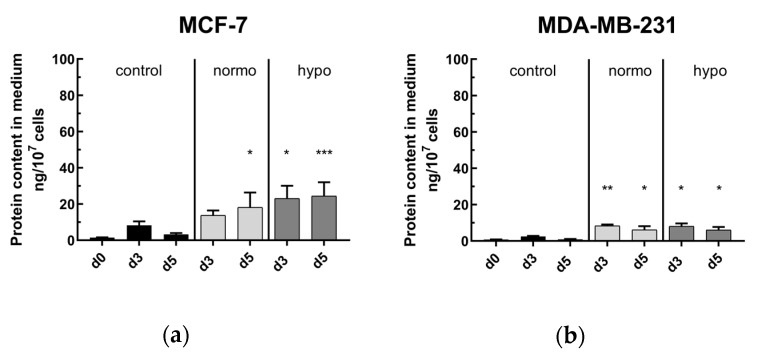
Analysis of danger signal HSP70 in the supernatant of (**a**) MCF-7 and (**b**) MDA-MB-231 breast cancer cells after normo- and hypofractionated radiotherapy. Cells were irradiated either by normofractionation (normo) at single doses of 2 Gy or by hypofractionation (hypo) at single doses of 5 Gy and analyzed on day 3 (d3) and day 5 (d5). The protein content of the danger signal HSP70 was analyzed by sandwich ELISA and is related to 1 × 10^7^ cells. Mean ± S.D. are presented from at least three independent experiments, each measured in duplicates. Significance test was conducted using Kruskal-Wallis test with uncorrected Dunn´s multiple comparison, by comparing the protein content of the respective irradiation treatment to the corresponding controls of mock-treated cells at the indicated time points (d3, d5); * (*p* < 0.1), ** (*p* < 0.01), *** (*p* < 0.001).

**Figure 5 cancers-12-01082-f005:**
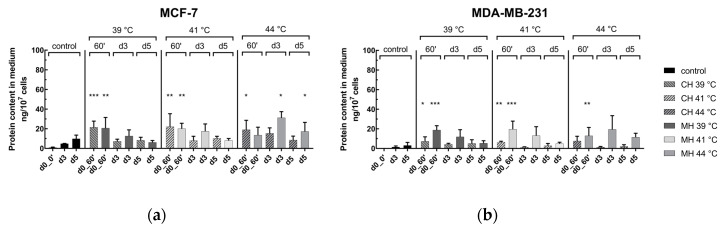
Analysis of danger signal HSP70 in the supernatant of (**a**) MCF-7 and (**b**) MDA-MB-231 breast cancer cells after conventional warm-water or microwave heating. Cells were heated either by conventional warm-water (CH) or microwaves (MH) within the self-designed hyperthermia system to three clinically relevant temperatures, i.e., 39 °C, 41 °C and 44 °C, for an effective time of 60 min (d0_0′ to d0_60). After the effective treatment time of 60 min, the cells were distributed into 75 cm^2^ T-flasks for analysis on day 3 (d3) and day 5 (d5). The protein content of the danger signal HSP70 was analyzed by sandwich ELISA and is related to 1 × 10^7^ cells. Mean ± S.D. are presented from at least three independent experiments, each measured in duplicates. Significance test was conducted using Kruskal-Wallis test with uncorrected Dunn´s multiple comparison, by comparing the protein content of the respective irradiation treatment to the corresponding controls of mock-treated cells at the indicated time points (d0_0′, d3, d5). The exemplary shown controls of d0_0′, d3 and d5 are from 39 °C and CH experiments and differ only slightly to other treatment controls; * (*p* < 0.1), ** (*p* < 0.01), *** (*p* < 0.001).

**Figure 6 cancers-12-01082-f006:**
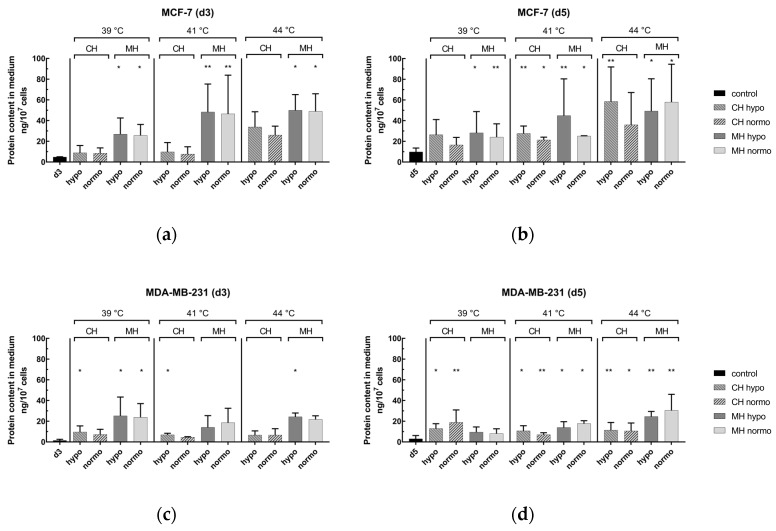
Analysis of danger signal HSP70 in the supernatant of (**a**,**b**) MCF-7 and (**c**,**d**) MDA-MB-231 breast cancer cells after combinatory treatment by conventional warm-water or microwave heating and radiotherapy on day 3 (**a**,**c**) as well as on day 5 (**b**,**d**). Cells were heated either by conventional warm-water (CH) or microwaves (MH) within the self-designed hyperthermia system to three clinically relevant temperatures, i.e., 39 °C, 41 °C and 44 °C, for an effective time of 60 min (d0_0′ to d0_60). After the effective treatment time of 60 min, the cells were distributed into 75 cm^2^ T-flasks for additional treatment, i.e., normofractionation (normo) at single doses of 2 Gy or by hypofractionation (hypo) at single doses of 5 Gy, and analysis on day 3 (d3) and day 5 (d5). The protein content of the danger signal HSP70 was analyzed by sandwich ELISA and is related to 1 × 10^7^ cells. Mean ± S.D. are presented from at least three independent experiments, each measured in duplicates. Significance test was conducted using Kruskal-Wallis test with uncorrected Dunn´s multiple comparison, by comparing the protein content of the respective irradiation treatment to the corresponding controls of mock-treated cells at the indicated time points (d3,d5). The exemplary shown controls of d3 and d5 are from 39 °C and CH experiments and differ only slightly to other treatment controls; * (*p* < 0.1), ** (*p* < 0.01).

**Figure 7 cancers-12-01082-f007:**
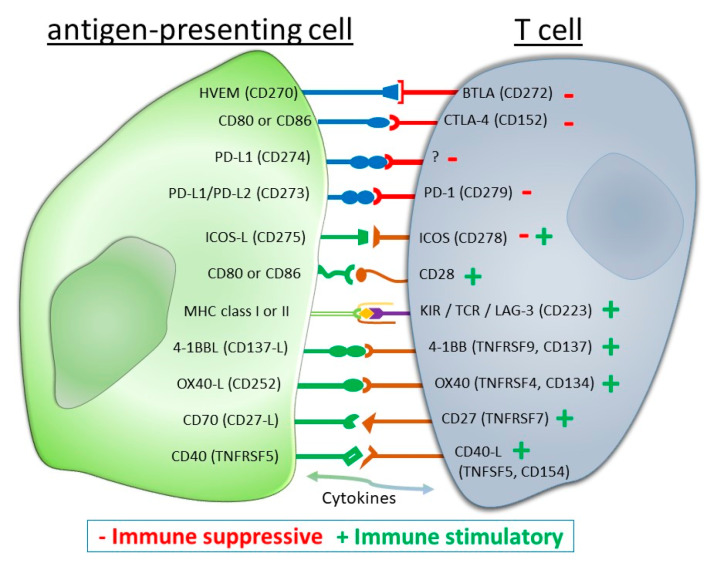
Immunological synapse between an antigen-presenting cell or a tumor cell and a T cell. Both, immune suppressive (−) and immune stimulatory (+) checkpoint molecules are present.

**Figure 8 cancers-12-01082-f008:**
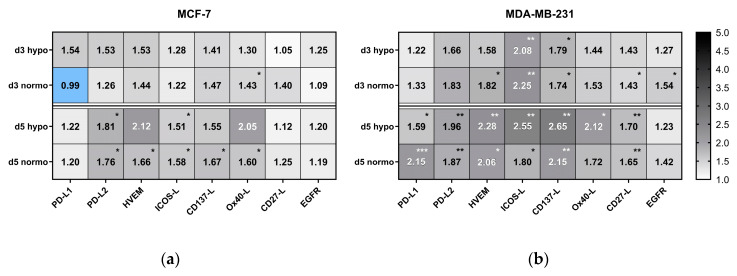
Heatmap of normalized expression (change in mean fluorescence intensity compared to mock-treated cells) of immune checkpoint molecules and of EGFR on day 3 (d3) and day 5 (d5) on the cell surface of (**a**) MCF-7 and (**b**) MDA-MB-231 breast cancer cells after normo- and hypofractionated radiotherapy. Significance test was conducted using Kruskal-Wallis test with uncorrected Dunn´s multiple comparison from three independent experiments, each measured in duplicates * (*p* < 0.1), ** (*p* < 0.01), *** (*p* < 0.001).

**Figure 9 cancers-12-01082-f009:**
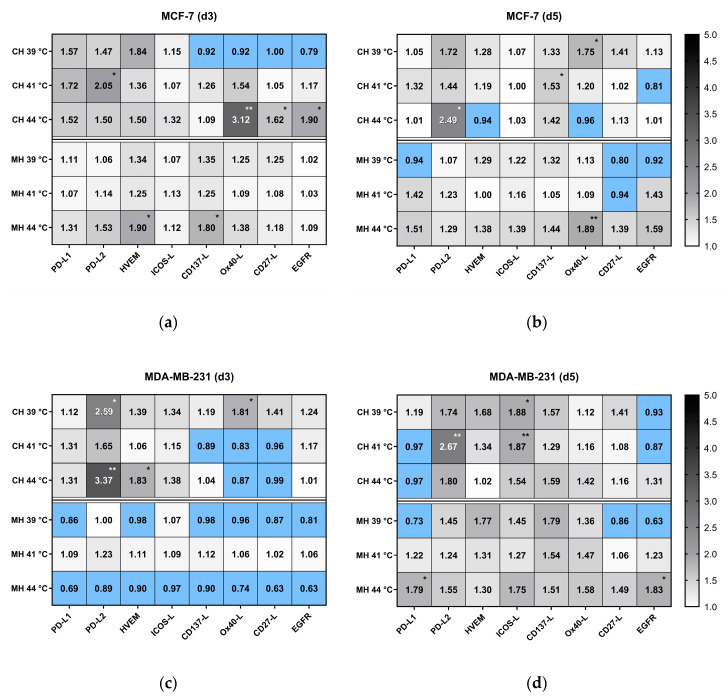
Heatmap of normalized expression (change in mean fluorescence intensity compared to mock-treated cells) of immune checkpoint molecules and of EGFR on the cell surface of (**a**,**b**) MCF-7 and (**c**,**d**) MDA-MB-231 breast cancer cells after conventional warm-water (CH) or microwave heating (MH) on day 3 (d3) (a and c) and day 5 (d5; b and d). Cells were heated either by CH or MH within the self-designed hyperthermia system to three clinically relevant temperatures, i.e., 39 °C, 41 °C and 44 °C, for an effective time of 60 min. Afterwards, the cells were distributed into 75 cm^2^ T-flasks for analysis on d3) and d5. Significance test was conducted using Kruskal-Wallis test with uncorrected Dunn´s multiple comparison from three independent experiments, each measured in duplicates. * (*p* < 0.1), ** (*p* < 0.01).

**Figure 10 cancers-12-01082-f010:**
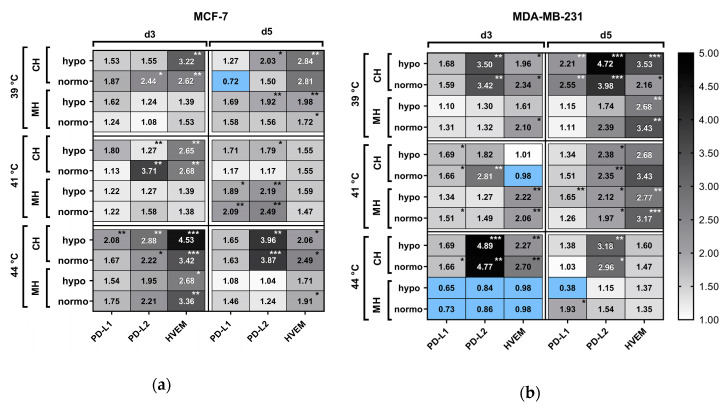
Heatmap of normalized expression (change in mean fluorescence intensity compared to mock-treated cells)) of immune suppressive immune checkpoint molecules on cell the cell surface of (**a**) MCF-7 and (**b**) MDA-MB-231 breast cancer cells after combinatory treatment of conventional warm-water (CH) or microwave heating (MH) and radiotherapy. Cells were heated either CH or MH within the self-designed hyperthermia system to three clinically relevant temperatures, i.e., 39 °C, 41 °C and 44 °C, for an effective time of 60 min. Afterwards, the cells were distributed into 75 cm^2^ T-flasks for additional treatment, i.e., by normofractionated (normo) RT at single doses of 2 Gy or by hypofractionated (hypo) RT at single doses of 5 Gy, and analysis on d3 and d5.Significance test was conducted using Kruskal-Wallis test with uncorrected Dunn´s multiple comparison from three independent experiments, each measured in duplicates. * (*p* < 0.1), ** (*p* < 0.01), *** (*p* < 0.001).

**Figure 11 cancers-12-01082-f011:**
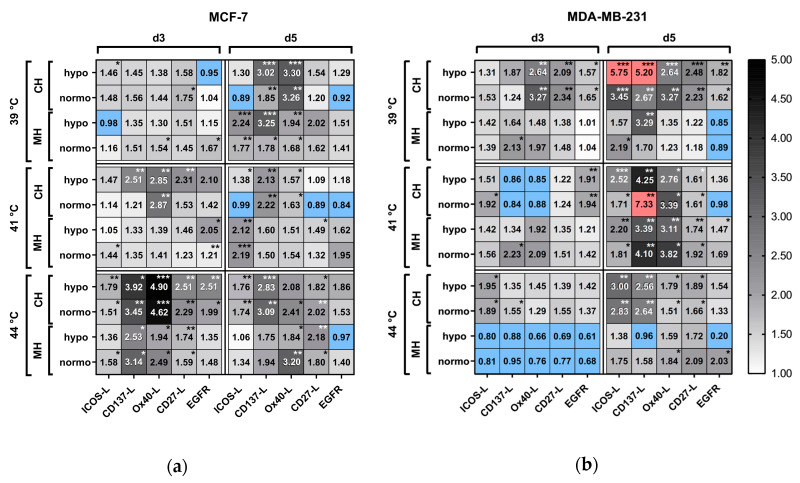
Heatmap of normalized expression (change in mean fluorescence intensity compared to mock-treated cells) of immune stimulatory checkpoint molecules and EGFR on the cell surface of (**a**) MCF-7 and (**b**) MDA-MB-231 breast cancer cells after combinatory treatment of conventional warm-water (CH) or microwave heating (MH) and radiotherapy. Cells were heated either by CH or MH within the self-designed hyperthermia system to three clinically relevant temperatures, i.e., 39 °C, 41 °C and 44 °C, for an effective time of 60 min. Afterwards, the cells were distributed into 75 cm^2^ T-flasks for additional treatment, i.e., by normofractionated (normo) RT at single doses of 2 Gy or by hypofractionated (hypo) RT at single doses of 5 Gy, and analysis on d3 and d5. Significance test was conducted using Kruskal-Wallis test with uncorrected Dunn´s multiple comparison from three independent experiments, each measured in duplicates. * (*p* < 0.1), ** (*p* < 0.01), *** (*p* < 0.001).

**Figure 12 cancers-12-01082-f012:**
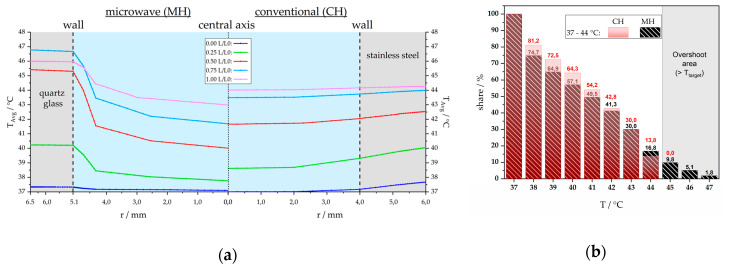
Simulated radial temperature profile and percentage of the respective temperature range: (**a**) fluid medium (blue field) for microwave (left, MH) and conventional (right, CH) heating at five normalized positions (L/L0) along the heating device. Process conditions in this example were T_target_ = 44 °C and ${\dot{V}}_{m}$ = 2 mL/s at P_MW_ = 100 W, respectively T_W_ = 48.5 °C. (**b**) Percentage of the respective temperature range with conventional heating (filled bars and percentage rates in red, CH) or microwave heating (dashed bars and percentage rates in black, MH). Additional information about the heating systems used for the experiments is provided in Section 4.1 and Section 4.2.

**Figure 13 cancers-12-01082-f013:**
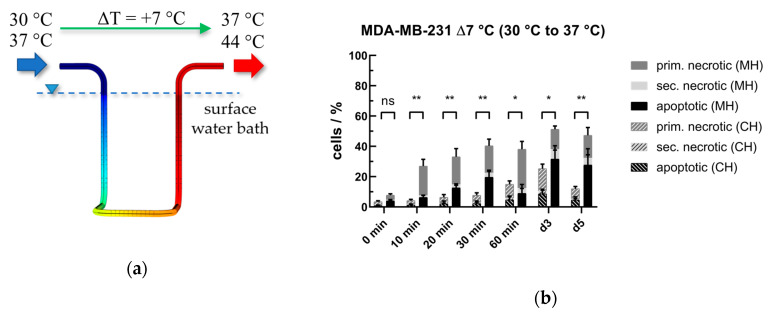
(**a**) Exemplary radial temperature profile for heating the cells by a ΔT of +7 °C and (**b**) cell death forms after conventional or microwave heating of MDA-MB-231 breast cancer cells from 30 °C to 37 °C. Significance test was conducted using Kruskal-Wallis test with uncorrected Dunn´s multiple comparison from three independent experiments, each measured in duplicates* (*p* < 0.1), ** (*p* < 0.01).

**Figure 14 cancers-12-01082-f014:**
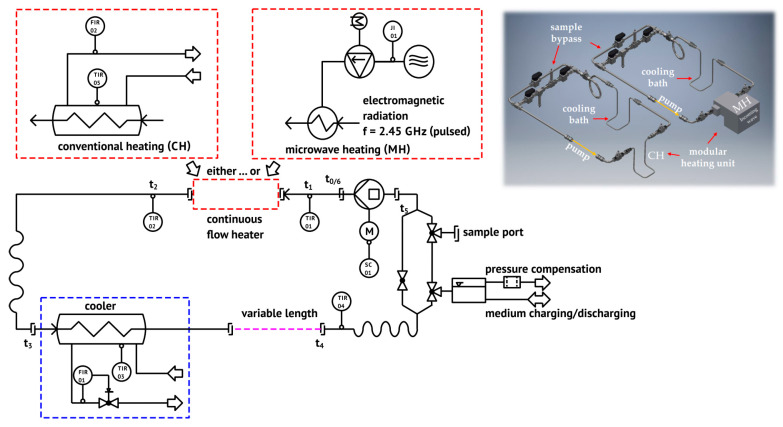
Flowchart of the self-designed, modular lab-scale closed loop system for the inactivation of tumor cell suspension under circulating flow with a modular heating unit: use of conventional warm-water heater (CH, left upper part of the flow chart) and microwave heater (MH, right upper part of the flow chart) and CAD drawing of both heating system (right upper part).

**Figure 15 cancers-12-01082-f015:**
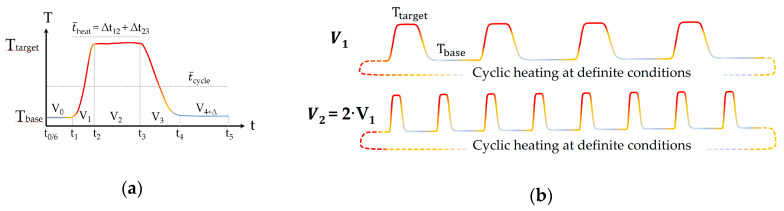
(**a**) Schematic view of one single heating cycle from *T*_base_ to *T*_target_ and back to *T*_base_ and (**b**) illustration of periodic heating cycle with two different flow rates but identical treatment time, *t*_eff_, of the heating system described in Figure 14.

**Figure 16 cancers-12-01082-f016:**
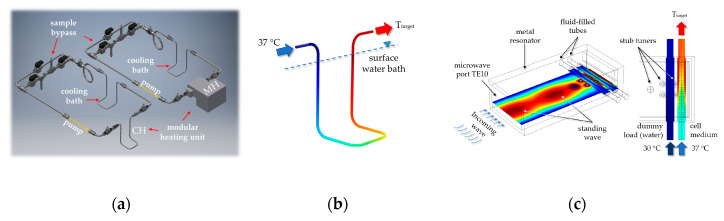
(**a**) CAD drawing of the closed-loop heating system with regions of interest, i.e., modular heating unit CH or MH (see also Figure 14 and (**b**) conventional warm-water and (**c**) microwave heating model for numerical simulations and modeling.

**Figure 17 cancers-12-01082-f017:**
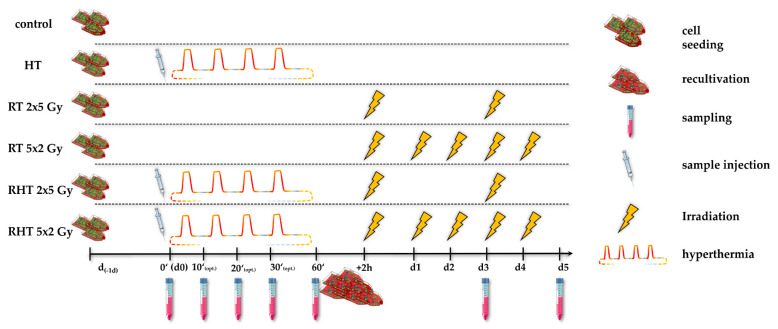
Experimental treatment set-up: The cells were seeded cell line-dependent one day prior to the treatment (d_-1d_) for not more than 90 % of confluency during the whole treatment, e.g., 3 × 10^5^ cells per 75 cm^2^ t-flask of MDA-MB-231 for sampling on day 5 in the control arm. Standard sampling in all arms was performed on day 0 (d0), d3 (72h) and d5 (120h). In the HT and combinatory arm, i.e., radiotherapy & hyperthermia (RHT), sample injection of 1x10^7^ cells into the HT system was done at time point d0, followed by an effective heating session of at most 60 min. After the HT treatment, cells were recultivated into 75 cm^2^ t-flasks according to their subsequent treatment. Irradiation in the radiotherapy arm (RT) and combinatory RHT arm was performed in 75 cm^2^ t-flasks with clinically relevant doses of either 2 × 5 Gy or 5 × 2 Gy. Irradiation in the respective RT and RHT arms was always performed at the same time.

**Figure 18 cancers-12-01082-f018:**
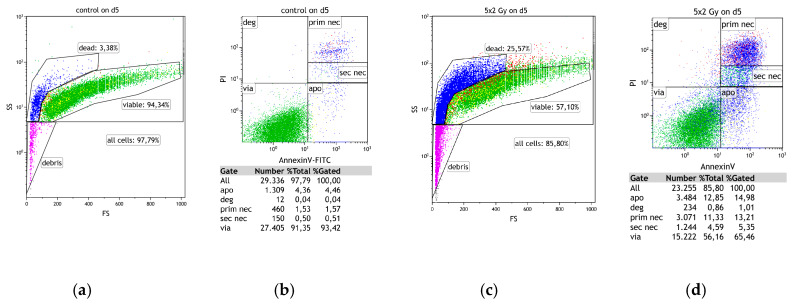
Cell death forms detection by Annexin V/PI staining and analysis by flow cytometry. Morphological changes and AnnexinV/PI staining of MDA-MB-231 tumor cells without (**a**,**b**) and with irradiation at 5 × 2 Gy (**c**,**d**) at day 5 after treatment is displayed exemplarily. Flow cytometry was used to distinguish viable and dead cells (**a**,**c**) by their FSc/SSc morphology as well as by their AnnexinV/PI binding properties (**b**,**d**); via: viable; apo: apoptotic; prim. nec: primary necrosis; sec. nec: secondary necrosis; d5: day 5.

**Table 1 cancers-12-01082-t001:** Antibodies used for the analyses of the surface expression of immune checkpoint molecules and of EGFR by multicolor flow cytometry.

Marker	Laser Color	Manufacturer	Cat. Nr.	µL Per Well
PD-L1 (CD 274)	BV 605	BioLegend	329724	0.5 ^1^
PD-L2 (CD 273)	APC	BioLegend	345508	0.5 ^1^
EGF-Receptor	PE	BioLegend	352904	0.5 ^1^
ICOS-L (CD 275)	BV 421	BD Bioscience	564276	0.5 ^1^
HVEM (CD 270)	APC	BioLegend	318808	0.5 ^2^
OX40-L (CD252)	PE	BioLegend	326308	0.5 ^2^
TNFRSF9 (CD137-L)	BV 421	BioLegend	311508	0.5 ^2^
CD27-L (CD70)	FITC	BioLegend	355106	0.5 ^2^
Zombie NIR	APC-A750	BioLegend	423105	0.1 ^1,2,3^
FACS-buffer	2% FBS in DPBS (sterile)			97.9 ^1,2^/99.9 ^3^

^1^ Mastermix #1 stained; ^2^ Mastermix #2 stained; ^3^ unstained Mix for #1 and #2.

**Table 2 cancers-12-01082-t002:** Complex relative permittivity $\varepsilon_{r}=\varepsilon_{r}^{\prime }-j\varepsilon_{r}^{\prime \prime }$ of water at 2.45 GHz in dependence of the temperature.

Parameter	25 °C	35 °C	45 °C	55 °C
$\varepsilon_{r}^{\prime }\;\left(T\right)$	76.7	74.0	70.7	67.5
$\varepsilon_{r}^{\prime \prime }\;\left(T\right)$	12.04	9.398	7.494	6.008

**Table 3 cancers-12-01082-t003:** Temperature-invariant material properties necessary for the simulation of the microwave (MH) and warm-water bath (CH) models (see Figure 16).

Parameter	Stainless Steel (CH)	Stainless Steel (MH)	Quartz Glass (MH)	Air (MH & CH)
Relative permeability, *µ*_r_ (-)	1	1	1	1
Electric conductivity, *σ* (S/m)	—	4 · 10^6^	—	0.85
Relative permittivity, ε_r_ (-)	—	—	3.78 - *j*2 · 10^−4^	1
Density, ρ (kg/m³)	7800	7850	2200	1.2
Heat conductivity, *λ* (W/mK)	15	44.5	1.1	0.026
Heat capacity, *c*_p_ (J/kgK)	420	475	480	1006

**Table 4 cancers-12-01082-t004:** Parameter levels for heat treatment (HT&RHT) of breast cancer cells.

Parameter	Levels	Unit
T_target_	37.0, 39.0, 41.0, 44.0	°C
$\dot{V}$ _m_	2.0	mL/s
*t*_eff_ *	10 ^1^, 20 ^1^, 30 ^1^, 60	min
normo(fractionation) ^2^	5 fractions of 2 Gy consecutively	
hypo(fractionation) ^2^	2 fractions of 5 Gy on d0 and d3	

* effective heating time is ¼ of the total cyclic heating time, ^1^ this could be done optionally, ^2^ in case of RHT, first irradiation was done 2 h after hyperthermia heating.

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
