# Peer review of "Differences of the Immune Phenotype of Breast Cancer Cells after Ex Vivo Hyperthermia by Warm-Water or Microwave Radiation in a Closed-Loop System Alone or in Combination with Radiotherapy"

_cancers, 2020, doi:10.3390/cancers12051082_

Round 1
Reviewer 1 Report
Hader et al. created a new in vitro experimental system and reported that adding hyperthermia to radiotherapy altered the expression of molecules involved in the immune response. Because their study is well-designed and the data are robust, this reviewer thinks that the paper could be accepted without major modification. However, this reviewer would like the authors to describe if a clinical application is considered, how should hyperthermia be applied to the treatment of metastatic lesions deep in the body, such as the lungs and liver.
Author Response
Thank you for taking the time to review our paper thoroughly, for the positive feedback, and for the constructive comments.
A clinical application as described in our study, i.e. a closed-loop system, is not considered to be used for clinical hyperthermia treatments of superficial or deep tumors. We apologize for having this not pointed out clear enough. But, the described model system allows for the first time to analyze the impact of microwave heating on the immunogenicity of tumor cells by avoiding generation of temperature hot spots. As just recently summarized by Paulides et al. (PMID: 32229271), in particular microwave and radiofrequency hyperthermia devices provide a variety of quality-controlled heating approaches that allow to treat most of solid cancers regardless of the size. By these application forms, adding hyperthermia to conventional radio(chemo)therapy was shown to be beneficial for the clinical outcome in the context of randomized phase I-III trials. However, one has to carefully consider in the clinics which tumor can be heated and which one not. Hyperthermia of metastatic lesions deep in the body, such as lung and liver, are not easy to perform by current available hyperthermia systems. Two main reasons are the high heat dissipation in these well-perfused tissue entities and (respiratory) motion and air which affects invasive targeted temperature measurement and makes adequate heat delivery difficult, respectively. However, improvements in treatment planning software (segmentation, tissue bioheat properties etc.) and non-invasive real-time temperature monitoring/adaption might bring improvements to achieve the desired temperature elevation even in such tumor entities in the future (PMID: 32229271).
Additionally, our closed-loop system could be used in the future to generate whole tumor cell-based vaccines based as conceptual described in detail by Seitz et al. (PMID: 31555582).
We added the following to the revised version of our manuscript:
Introduction:
Hyperthermia, i.e. local heating of tumor tissue to temperatures between 39 °C to 45 °C for around 60 minutes, has been proven to enhance radiation and chemotherapeutic effects [9,10]. As just recently summarized by Paulides et al. [11], in particular microwave and radiofrequency HT devices provide a variety of quality-controlled heating approaches that allow to treat most of solid cancers regardless of the size. By these application forms, adding HT to conventional radio(chemo)therapy was shown to be beneficial for the clinical outcome in the context of randomized phase I-III trials. However, one has to carefully consider in the clinics which tumor can be heated and which one not. HT of metastatic lesions deep in the body, such as lung and liver, are not easy to perform by current available HT systems. Two main reasons are the high heat dissipation in these well-perfused tissue entities and (respiratory) motion and air which affects invasive targeted temperature measurement and makes adequate heat delivery difficult, respectively. However, improvements in treatment planning software (segmentation, tissue bioheat properties etc.) and non-invasive real-time temperature monitoring/adaption might bring improvements to achieve the desired temperature elevation even in such tumor entities in the future [11].
It is further accepted that HT impacts the immune system [12–14]. For recurrent breast cancer, HT has become a well-accepted therapy when being applied in combination with RT [14,15]. Nevertheless, little is known how HT particularly impacts the immunogenicity of breast cancer cells, whether temperature-dependencies do exist and if heating methods differ from each other. Willerding et al. investigated the temperature characteristics in tumor-bearing rats by three different heating techniques, i.e. lamp, laser and warm-water bath. One of their key results regarding the different heating technique was that intratumoral temperature increase and distribution differed significantly from each other, also in dependence of the tumor size [16]. In this context, in clinical trials, HT is predominantly delivered via radiative/microwave applicators [9] with quality assurance guidelines for superficial [17] and interstitial hyperthermia [18]. In contrast to that, pre-clinical hyperthermia lacks standardized treatment guidelines and systems, that is why experiments are mostly based on incubators or warm-water baths. Our described model system allows the first time to analyze the impact of microwave heating on the immunogenicity of tumor cells by avoiding generation of temperature hot spots. For translation of pre-clinical findings into clinical trials, it is important to use similar settings and parameters as much as possible. Immune checkpoint inhibitors were clinically approved for the treatment of different solid tumor entities, but the response rates are still below 15 % [19]. This raises questions of why so many patients do not respond to immunotherapy, and how can standard cancer therapy methods such as RT contribute to increase the number of responders. HT could be integrated in such multimodal settings as additional immune modulator [13].
Discussion:
Additionally, our closed-loop system could be used in the future to generate whole tumor cell-based vaccines based on the concept that was just recently described by Seitz et al. [76].
Reviewer 2 Report
1 Introduction:
Most of the introduction is on possible contribution of hyperthermia to immunogenic effects of radiotherapy. In the very last paragraph of the introduction a seemingly entirely different subject of differences between waterbath heating and EM heating is mentioned, and the connection between the different modes of heating and the previous section on immunology is not thoroughly explained. The authors do point out that preclinical research generally uses water bath heating, and clinical application microwave heating, but that would only strike me as relevant if there were a mismatch between preclinical and clinical results, and the authors do not mention that.
4 Materials and methods:
The setup for the preclinical microwave system shown in figure 14 is quite novel, but an intriguing part is that this form of implementing microwave heating is different from the implementation in clinical systems in the sense that a traditional clinical microwave hyperthermia treatment involves a continous period of 60 minutes of heating, whereas in this set up the cell suspension is undergoing many cycles of rapid heating, a short period at the designated elevated temperature, followed by cooling back to the control temperature. This cyclic heating/cooling is done for both the water bath and microwave heating so these preclinical results can be compared to each other, but comparison with clinical practice is more challenging. And I have been searching the manuscript to find the details on the time points t1, t2, t3, t4 and t5 determining the cycle parameters as mentioned in the caption of figure 15, and formula (1), but I could not find the specific numbers, also not in section 4.4 where you are referring to in line 616. All I can read in table 3 in section 4.4 is that the effective time at the elevated temperature was 60 minutes, but not the overall time needed to achieve that 60 minutes of effective heating and also not how many cycles were required to arrive at those 60 minutes. And I think that those are quite relevant parameters for understanding the factual heating protocol as well as interpreting the outcome. And the absence of these data is strange in view of the abundant number of data presented on other parameters in the manuscript.
Figure 17 shows the overall set up with the different combinations of hyperthermia and radiotherapy. I see that first hyperthermia is given, followed two hours later by radiotherapy, either in a 2x5Gy or a 5x2Gy schedule with the first radiotherapy session on the day of the hyperthermia treatment. Why did you select a 2 hour time interval? You are making remark on line 492 that the clinical interval between radiation therapy and HT is not precisely defined, but I missed a justification for that choice of a two hour interval in your experiments. The choice of interval may affect the synergy between radiotherapy and hyperthermia. In the context of this manuscript the mutual effects of hyperthermia and radiotherapy are mainly used as additive effects, which can make sense with a focus on exploring the immunogenic potential of either radiotherapy and of microwave based versus water bath induced hyperthermia.
The results in Figure 2 are quite interesting and show the effect of microwave versus water bath and also of the temperature levele achieved with microwave heating, where if I am correct the same power is used for the different temperature levels which would suggest that the effects found are due to a combination of temperature and the fact that cells were exposed to microwave radiation.
Figure 3 is also interesting, this time also the water bath heating seems to have an effect.
The HSP-70 induction seems tob e quite dependent on the heating method, and on the cell line. Figure 5 shows quite erratic results but on the whol with a more significant effect for microwave heating and at higher temperatures, even after 5 days.
You are making a statement in section 3.2 on the possible effect of different temperature levels on the immunomodulation, but I am missing a link in this section with your own results achieved. Do your results suggest an idea whether there is a specific dose-effect relationship?
Summarizing, these experiments are quite interesting, but I am missing a number of crucial details in the description of the experiments, as well as some elements in the discussion, particularly where the preclinical experiments deviate from what is normal in clinical and in most preclinical protocls with this particalar cyclic form of heating. Please discuss the impact of those differences. That being said, I do feel this is a very interesting paper and a first step in better understanding the benefits of different forms of hyperthermia in combatting cancer.
